



# Photolysis and oxidation by OH radicals of two carbonyl

2  # nitrates: 4-nitrooxy-2-butanone and 5-nitrooxy-2-pentanone

Bénédicte Picquet-Varrault[1], Ricardo Suarez-Bertoa[2], Marius Duncianu[3], Mathieu Cazaunau[1],

4  Edouard Pangui[1], Marc David[1], Jean-François Doussin[1]

[1] LISA, UMR CNRS 7583, Université Paris-Est Créteil, Université de Paris, Institut Pierre Simon Laplace

6  (IPSL), Créteil, France

[2] European Commission Joint Research Centre (JRC), Ispra, Italy

8  [3] System Analyst Interscience BV, Brussels, Belgium

10  *Correspondence to*: B. Picquet-Varrault (benedicte.picquet-varrault@lisa.u-pec.fr)

12  **Abstract**

Multifunctional organic nitrates, including carbonyl nitrates, are important species formed in NOx

14  rich atmospheres by the degradation of VOCs. These compounds have been shown to play a key role

in the transport of reactive nitrogen and consequently in the ozone budget, but also to be important

16  components of the total organic aerosol. However, very little is known about their reactivity in both

gas and condensed phases. Following a previous study we published on the gas-phase reactivity of α-

18  nitrooxy ketones, the photolysis and the reaction with OH radicals of 4-nitrooxy-2-butanone and 5-

nitrooxy-2-pentanone, respectively a β-nitrooxy ketone and a γ-nitrooxy ketone, were investigated for

20  the first time in simulation chambers. Ambient photolysis frequencies calculated for 40° latitude

North were found to be $(4.2 \pm 0.6) \times 10^{-5}$ s$^{-1}$ and $(2.2 \pm 0.7) \times 10^{-5}$ s$^{-1}$ for 4-nitrooxy-2-butanone and

22  5-nitrooxy-2-pentanone, respectively. These results demonstrate that photolysis is a very efficient

sink for these compounds with atmospheric lifetimes of few hours. It was also concluded that,

24  similarly to α-nitrooxy ketones, β-nitrooxy ketones have enhanced UV absorption cross sections and

quantum yields equal or close to unity. γ-nitrooxy ketones have been shown to have lower

enhancement of cross sections which can easily be explained by the increasing distance between the

two chromophore groups. Thanks to a products study, branching ratio between the two possible

photodissociation pathways are also proposed. Rate constants for the reaction with OH radicals were

found to be $(2.9 \pm 1.0) \times 10^{-12}$ cm$^3$ molecule$^{-1}$ s$^{-1}$ and $(3.3 \pm 0.9) \times 10^{-12}$ cm$^3$ molecule$^{-1}$ s$^{-1}$,

respectively. These experimental data are in good agreement with rate constants estimated by the

SAR of Kwok and Atkinson (1995) when using the parametrization proposed by Suarez-Bertoa et al.

(2012) for carbonyl nitrates. Comparison with photolysis rates suggests that OH-initiated oxidation of

carbonyl nitrates is a less efficient sink that photodissociation but is not negligible in polluted area.

**1 Introduction**



Organic nitrates (ONs) play an important role as sinks or temporary reservoirs of NOx, as well as on ozone
production in the atmosphere (Perring et al., 2013; Perring et al., 2010; Ito et al., 2007). They are formed by the
degradation of VOCs in NOx rich air (typically urban areas) through two main processes: i) the reaction of
38 peroxy radical, produced by the oxidation of VOCs, with NO. The major pathway is generally the reaction (1a)
that leads to $NO_2$ formation. The reaction (1b) is a minor channel but it becomes gradually more important with
40 increasing peroxy radical carbon chain length (Atkinson and Arey, 2003; Finlayson-Pitts and Pitts, 2000).

$$RO_2 + NO \rightarrow RO + NO_2 \qquad\qquad (1a)$$

$$RO_2 + NO \rightarrow RONO_2 \qquad\qquad (1b)$$

ii) The reaction of unsaturated VOCs with $NO_3$ radical, which proceeds mainly by addition of the nitrate radical
on the double bond to produce nitro-alkyl radicals that can evolve into organic nitrates.

Among the organic nitrates, a variety of multifunctional species such as hydroxy-nitrates, carbonyl-nitrates and
46 dinitrates are formed. The formed species have been shown to significantly contribute to the nitrogen budget in
both rural and urban areas (Perring et al. 2013). Beaver et al. (2012) have observed that carbonyl nitrates, formed
as second generation nitrates from isoprene, are an important fraction of the total organic nitrates observed over
Sierra Nevada in summer. These observations are supported by several studies that investigated the
50 photooxidation of isoprene in simulation chambers (Paulot et al., 2009, Müller et al., 2014). These
multifunctional organic nitrates are also semi-/non-volatile and highly soluble species and are thus capable of
52 partitioning into the atmospheric condensed phases (droplets, aerosols). Numerous field observations of the
chemical composition of atmospheric particles have shown that organic nitrates represent a significant fraction
(up to 75% in mass) of the total organic aerosol (OA) demonstrating that these species are important components
of total OA (Ng et al., 2017).

Several modeling studies have also confirmed that multifunctional organic nitrates, in particular isoprene
nitrates, play a key role in the transport of reactive nitrogen and consequently in the formation of ozone and
58 other secondary pollutants at the regional and global scales (Horowitz et al., 2007; Mao et al., 2013; Squire et al.,
2015). In particular, Mao et al. (2013) have performed simulations based on data from the ICARTT aircraft
campaign across the eastern U.S. in 2004. They have shown that ONs, which are mainly composed of secondary
organic nitrates, including a large fraction of carbonyl nitrates, provide an important pathway for exporting NOx
from the U.S.'s boundary layer, even exceeding the export of PANs. However, these modeling studies also point
out the need for additional experimental data to better describe the sinks in both gas and condensed phases of
64 multifunctional organic nitrates in models.

Recent experimental studies have revealed that hydrolysis in aerosol phase may be a very efficient sink of
66 organic nitrates in the atmosphere (Bean and Hildebrand Ruiz, 2015; Rindelaub et al., 2015). These works also
suggest that the rate of these reactions strongly depends on the organic nitrate chemical structure and that
additional works are needed to better understand these processes. In the gas-phase, photolysis and reaction with
OH radical are expected to dominate the fate of organic nitrates (Roberts et al., 1990; Turberg et al., 1990). In a
70 previous study, we have measured the photolysis frequencies and the rate constants for the OH-oxidation of 3
carbonyl nitrates (α-nitrooxyacetone, 3-nitrooxy-2-butanone, and 3-methyl-3-nitrooxy-2-butanone) and we have
72 shown that photolysis is the dominant sink for these compounds (Suarez-Bertoa et al., 2012). By comparison



with absorption cross sections provided by Barnes et al. (1993), Müller et al. (2014) suggested i) that the α-nitrooxy ketones have enhanced absorption cross sections, due to the interaction between the –C=O and the -ONO$_2$ chromophore groups and ii) that the quantum yield is close to unity photolysis and O–NO$_2$ dissociation is the likely major channel. They also showed that this enhancement was larger at the higher wavelengths, where the absorption by the nitrate chromophore is very small. Therefore, they concluded that the absorption by the carbonyl chromophore was the one enhanced due to the neighbouring nitrate group.

These results are significant as they demonstrate that photolysis rates of these multifunctional species cannot be calculated as the sum of the monofunctional species (ketone + alkyl nitrate) ones. However, only α-nitrooxy ketones were studied in those works which leaves open the question of the persistence of the enhancement effect when the distance between the two functional groups increases. More recently, Xiong et al. (2016) have studied the photochemical degradation (photolysis, OH-oxidation and ozonolysis) of *trans*-2-methyl-4-nitrooxy-2-buten-1-al (also called 4,1-isoprene nitrooxy enal) in order to better assess – as a model compound - the reactivity carbonyl nitrates formed by the NO$_3$-initiated oxidation of isoprene. This compound has a conjugated chromophore –C=C-C=O in β position of the nitrate group. The authors measured the absorption cross sections of the nitroxy enal and compared them to those for the monofunctional species, i.e. methacrolein and isopropyl nitrate. They concluded that molecules containing β-nitrooxy ketones functionalities have also enhanced UV absorption cross sections. They also studied the kinetic and mechanisms for the oxidation of the nitroxy enal by OH and O$_3$ and showed that the reaction with OH radicals is fairly fast. Photolysis and reaction with OH are thus the two main loss process *trans*-2-methyl-4-nitrooxy-2-buten-1-al leading to a tropospheric lifetime of less than 1 h.

Given the large contribution of the carbonyl nitrates to the organic nitrate pool and the importance of their photochemistry for the NOx budget, we present a study that aims at providing new experimental data on the gas-phase reactivity of these compounds. The study also seeks to disclose how photolysis and reaction with OH radical of carbonyl nitrates are affected by modifying their carbon chain length and the position of the two functional groups present in their molecular structure. Here, we provide the first photolysis frequencies and also the first rate constants for the OH-oxidation of two carbonyl nitrates: 4-nitrooxy-2-butanone and 5-nitrooxy-2-pentanone.

## 2 Experimental section

### 2.1 Reactants syntheses

As a usual OH precursor in simulation chamber experiments, isopropyl nitrite was synthesized by dropwise addition of a dilute solution of H$_2$SO$_4$ into a mixture of NaNO$_2$ and isopropanol following the classical protocol proposed by Taylor et al. (1980).

On the contrary, 4-nitroxy-2-butanone and 5-nitroxy-2-pentanone were synthesized for the first time. Because of their importance for atmospheric chemistry, a great care was taken to the development of a robust process: 4-nitroxy-2-butanone and 5-nitroxy-2-pentanone syntheses are based on the Kames' method (Kames et al., 1993). This method consists in a liquid/gas phase reaction where the corresponding hydroxy-ketone reacts with NO$_3$



radicals released from the dissociation of $N_2O_5$. $N_2O_5$ was preliminarily synthesized in a vacuum line by reaction

110 of $NO_2$ with ozone, as described by Scarfogliero et al. (2006). The synthesis of the carbonyl nitrate is performed in a dedicated vacuum line connected to two bulbs, one containing the hydroxy-ketone, the other one containing

112 $N_2O_5$. The two bulbs are also connected to each other. In a first step, the bulbs are placed in a liquid $N_2$ cryogenic trap and pumped in order to remove air and impurities. In a second step, the cryogenic trap is removed

114 from the bulb containing $N_2O_5$ in order to let it warm and transfer into the bulb containing the hydroxy-ketone. Then, the bulb containing both reactants is stirred and kept at ice temperature for approximatively 1h. Finally,

116 the resulting carbonyl nitrate and nitric acid, its co-product, were separated by liquid-liquid extraction using dichloromethane and water. The carbonyl nitrate structure and purity were verified by FT-IR and GC-MS.

118 Traces of impurities (HCOOH and $CH_3COOH$) have been detected. The carbonyl nitrates were stored at -18° C and under nitrogen atmosphere to prevent them from decomposition. Infrared spectra of 4-nitroxy-2-butanone

120 and 5-nitroxy-2-pentanone are available on EUROCHAMP Data Centre (https://data.eurochamp.org).

**2.2 Determination of photolysis frequencies**

122 The photolysis frequencies of the two carbonyl nitrates were determined by carrying out experiments in the CESAM simulation chamber which is only briefly described here as detailed information can be found in Wang

124 et al. (2011). The chamber consists of a 4.2 $m^3$ stainless steel vessel equipped with a multiple reflection optical system interfaced to a FTIR spectrometer (Bruker Tensor 37) and also with NO, $NO_2$ and $O_3$ analyzers (Horiba)

126 to monitor the composition of the gas phase. The chamber is also equipped with three high pressure xenon arc lamps (MH-Diffusion, MacBeam 4000) which provide a very realistic actinic flux (in comparison to the solar

128 one) allowing measuring photolysis frequencies under realistic conditions (Wang et al., 2011; Suarez-Bertoa et al., 2012). The intensity of the actinic flux was determined by measuring the photolysis rate of $NO_2$ ($J_{NO2}$) during

130 dedicated experiments. Hence, 400 ppbv of $NO_2$ in 1000 mbar of $N_2$ were injected into CESAM chamber and kept in the dark for 20 min. The lights were then turned on during 20 min, and finally the mixture was left in the

132 dark for an additional 20 min period. The photolysis frequency was subsequently determined using a kinetic numeric model developed for previous NOx photo-oxidation experiments in CESAM (Wang et al., 2011). The

134 fitting of modeled values from the measured data provided a $NO_2$ photolysis frequency equal to $3.0 \times 10^{-3}$ $s^{-1}$ (±0.01, 2σ error). Since the $J_{NO2}$ on July 1 at noon at 40° latitude North (overhead ozone column 200, albedo 0.1;

136 TUV NCAR Model, http://www.cprm.acd.ucar.edu/Models/TUV/Interactive_TUV/) is estimated to be equal to $9.6 \times 10^{-3}$ $s^{-1}$ and considering how close from the real sunlight is the irradiation light in CESAM, we assumed

138 that a proportional factor of 3.2 can be directly applied.

During a typical experiment, carbonyl nitrates were introduced into the chamber which was preliminarily filled

140 at atmospheric pressure with $N_2/O_2$ (80/20). For the injection, the bulb containing the carbonyl nitrate was connected to the chamber and slightly heated while it was flushed with $N_2$. Mixing ratios of carbonyl nitrates

142 ranged from hundreds ppb to ppm. Because carbonyl nitrates may decompose during the injection, large amounts of $NO_2$ (hundreds ppb) were present in the mixture. Cyclohexane was also added to the mixture as an OH-

144 scavenger with mixing ratios of approx. 4 ppm. Considering the fact that cyclohexane is much more reactive with OH radicals than ketonitrates (see section results), it was estimated that more than 95% of the OH radicals

146 were scavenged. The mixture was kept under dark conditions during two hours to be able to assess the impact of the reactor's walls and to minimize their effects by passivation. Then, the mixture was irradiated during 3 hours.





For most experiments, the mixture was finally left in the dark for approximately 1 hour after the irradiation period to allow for verifying if wall losses were constant during the entire duration of the experiment.

During the experiment, the carbonyl nitrate loss processes can be described as:

nitrate (walls) $\rightarrow$ products       (k)                 (2a)

nitrate + hv $\rightarrow$ products       (J)                 (2b)

$$-\frac{d[nitrate]}{dt} = (J + k) \times [nitrate] \qquad \text{(Eq. 1)}$$

$ln[nitrate]_t = ln[nitrate]_0 - (k + J) \times t$            (Eq. 2)

The reaction (2a) had to be added to the system to take into account the interaction or adsorption of the carbonyl
nitrates on the stainless steel walls of CESAM during the experiments. By plotting $ln[nitrate]_t$ *vs.* time, where $[nitrate]_t$ is the concentration of the carbonyl nitrate at time t, a straight line is obtained with a slope of (k + J).
The same approach was applied to each of the 'dark' periods, before and after irradiation, to determine their respective nitrate decay rates, namely $k_{before}$ and $k_{after}$ and k was calculated as the average of $k_{before}$ and $k_{after}$ for
each experiment. Finally, J was calculated as the difference between the loss rate during the irradiation period (k + J) and the averaged loss rate during the dark periods (k).

The uncertainties were calculated by adding the respective statistical errors ($2\sigma$) associated to the dark and light periods, the former set as the average of the uncertainties determined for both dark periods (i.e., before and after
irradiation). However, for some experiments, it was observed that the dark decay rate before irradiation was significantly higher than the one after, suggesting that wall losses may decrease with time due to a passivation
effect. In this case, the uncertainty was not calculated using the approach detailed above, the statistical error being too low compared to the difference between $k_{before}$ and $k_{after}$. The uncertainty was thus estimated in order to
include the lowest and the highest J values calculated with the highest and the lowest k value, respectively. Finally, the overall uncertainty associated with the photolysis rate of each of the carbonyl nitrates was calculated
as the average of the uncertainties obtained for each experiment, divided by the square root of the number of experiments (2 or 3).

**2.3 Determination of the OH-oxidation rate constants**

The kinetic experiments for the OH-oxidation of the carbonyl nitrates were performed in the CSA chamber at
room temperature and atmospheric pressure, in a mixture of $N_2/O_2$ (80/20). The chamber consists of a 977 L Pyrex$^{TM}$ vessel irradiated by two sets of 40 fluorescent tubes (Philips TL05 and TL03) that surround the
chamber. The emissions of these black lamps are centered on 360 and 420 nm, respectively. The chamber is equipped with a multiple reflection optical system with a path length of 180 m interfaced to a FTIR spectrometer
(Vertex 80 from Bruker). Additional details about this smog chamber are given elsewhere (Doussin et al., 1997; Duncianu et al., 2017).

The relative rate technique was used to determine the rate constant for the OH-oxidation of the carbonyl nitrates with methanol as reference compound. We used the IUPAC recommended value $k_{(methanol+OH)} = (9.0 \pm 1.8) \times$
$10^{-13}$ $cm^3$ $molecule^{-1}$ $s^{-1}$ (http://www.iupac-kinetic.ch.cam.ac.uk/). Hydroxyl radicals were generated by



photolyzing isopropyl nitrite. Initial mixing ratios of reactants (carbonyl nitrate, isopropyl nitrite, methanol and NO) were in the ppm range. As previously described, the carbonyl nitrate was introduced into the chamber by connecting the bulb to the chamber and by slightly heating and flushing it with $N_2$. NO was added to the mixture in order to enhance the formation of OH radicals by reaction with $HO_2$ radicals which are formed by isopropyl nitrite photolysis. All experiments were conducted during a 1 h period of continuous irradiation.

Prior to the experiments, it was verified that photolysis and wall losses of the studied compounds were negligible under our experimental conditions. This can be explained by the facts that i) the irradiation system of CSA chamber emits photons at significantly higher wavelengths than the one of CESAM chamber, and ii) the walls of the chamber are made of Pyrex which is more "chemically inert" than stainless steel. It was therefore assumed that reaction with OH is the only fate of both, the studied compound (nitrate) and the reference compound (methanol) and that neither of these compounds is reformed at any stage during the experiment. Based on these hypotheses, it can be shown that (Atkinson, 1986):

$$ln\frac{[nitrate]_0}{[nitrate]_t} = \frac{k_{nitrate}}{k_{methanol}} \times \frac{[methanol]_0}{[methanol]_t} \qquad\text{(Eq. 3)}$$

where $[nitrate]_0$ and $[methanol]_0$, and $[nitrate]_t$ and $[methanol]_t$ stand for the concentration of the carbonyl nitrate and the reference compound at times 0 and t, respectively. The plot $ln([nitrate]_0/[nitrate]_t)$ *vs.* $ln([methanol]_0/[methanol]_t)$ is linear with a slope equal to $k_{nitrate}/k_{methanol}$ and an intercept of zero. The uncertainty on $k_{nitrate}$ was calculated by adding the relative uncertainty corresponding to the statistical error on the linear regression ($2\sigma$) and the error on the reference rate constant (here 20 % for methanol).

**2.4 Chemicals and gases**

Dry synthetic air was generated using $N_2$ (from liquid nitrogen evaporation, >99.995% pure, <5 ppm $H_2O$, Linde Gas) and $O_2$ (quality N45, >99.995% pure, <5 ppm $H_2O$, Air Liquide). Chemicals obtained from commercial sources are: NO (quality N20, >99% Air Liquide), $NO_2$ (quality N20, >99% Air Liquide), 4-hydroxy-2-butanone (95% Aldich), 5-hydroxy-2-pentanone (95% Aldich), cyclohexane (VWR), methanol (J.T. Baker), $H_2SO_4$ (95% VWR), $NaNO_2$ (≥99 Prolabo), isopropanol (VWR).

**3 Results and discussion**

**3.1 Photolysis of carbonyl nitrates**

Photolyses of 4-nitrooxy-2-butanone and 5-nitrooxy-2-pentanone were investigated for the first time. Photolysis frequencies were determined by measuring the decay of the carbonyl nitrates in CESAM chamber under irradiation using a multipath FT-IR spectrometer. Figure 1 presents the kinetic plots obtained for the two compounds, where ln[nitrate] were plotted as a function of time. As explained in the experimental section, carbonyl nitrates were kept in the dark before and after the irradiation period in order to determine their decay rates due to wall losses. In this figure, a significant decrease was observed before and after irradiation for both compounds, suggesting that they adsorb or decompose on the walls. However, decay rates during irradiation periods are significantly faster than those in the dark showing that they are subject to additional loss by



photolysis. Photolysis frequencies were calculated as the difference between the decay rates in the dark and the one under irradiation. Results obtained for both compounds and for all experiments are given in Table 1. For 4-nitrooxy-2-butanone, photolysis frequencies are in good agreement despite the fact that decay rates in the dark differ from an experiment to another. For 5-nitrooxy-2-butanone, it can be seen that the decay rate in the dark before irradiation is significantly higher than the one after (in particular for experiments 3 and 4), suggesting that wall losses may decrease with time due to a passivation effect. Despite this, comparison of the three experiments showed good agreement.

The photolysis rates calculated for typical tropospheric irradiation conditions (see section 2.1) are $(4.2 \pm 0.6)\times 10^{-5}$ s$^{-1}$ for 4-nitrooxy-2-butanone and $(2.2 \pm 0.7)\times 10^{-5}$ s$^{-1}$ for 5-nitrooxy-2-pentanone. These results have been compared in Table 2 to those we obtained in a previous study (Suarez-Bertoa et al., 2012) for 3-nitrooxy-2-propanone, 3-nitrooxy-2-butanone and 3-methyl-3-nitrooxy-2-butanone using the same experimental approach. Experimental photolysis frequencies have also been compared to those calculated using cross sections published in the literature and by assuming a quantum yield equal to unity. Solar actinic flux was calculated from TUV NCAR model for the same conditions as those described in section 2.1. For 3-nitrooxy-2-propanone and 3-nitrooxy-2-butanone, cross sections were taken from Barnes et al. (1993). For 4-nitrooxy-2-butanone and 5-nitrooxy-2-pentanone, for which no data have been provided in the literature, cross sections were estimated using those for corresponding monofunctional species and by applying the enhancement factor ($r_{nk}$) obtained for 3-nitrooxy-2-propanone (Müller et al., 2014):

$$r_{nk} = \frac{s_{nk}}{s_n + s_k} \qquad\qquad\qquad\qquad (Eq.4)$$

Where $s_{nk}$, $s_n$ and $s_k$ are the absorption cross sections of the keto nitrate, the alkyl nitrate and the ketone, respectively. For 4-nitrooxy-2-butanone, cross sections of 2-butanone and 1-butyl nitrate were taken from IUPAC, 2006. For 5-nitroxy-2-pentanone, cross sections of 2-pentanone (http://satellite.mpic.de/spectral_atlas) and 1-pentyl nitrate (Clemitshaw et al., 1997) were used. From these results, it can be observed that the experimental photolysis frequencies ($J_{exp}$) obtained for 3-nitrooxy-2-propanone and 4-nitrooxy-2-butanone are very close and can be considered as equal within uncertainties. This suggests that the strong enhancement in the cross sections induced by the interaction between the two functional groups, which has been observed for α-nitrooxy ketones, also exists with the same amplitude for β-nitrooxy ketones. This is confirmed by the fact the experimental value for 4-nitrooxy-2-butanone is in very good agreement with the calculated value, obtained by assuming that the enhancement factor is the same as the one for 3-nitrooxy-2-propanone. This effect, nonetheless, is fading away when the two functions are one carbon further away: The experimental photolysis frequency obtained for 5-nitrooxy-2-pentanone is indeed significantly lower than those for 3-nitrooxy-2-propanone and 4-nitrooxy-2-butanone. It is also much lower than the J value calculated by assuming the same enhancement factor as for 3-nitrooxy-2-propanone. This result demonstrates that the enhancement is significantly reduced for γ-nitrooxy ketones even if it is probably not totally annihilated. This can easily be explained by the fact that the inductive effect of the nitrate group is expected to decrease when the distance between the functional groups increases. Finally, by comparing results for 3-nitrooxy-2-propanone, 3-nitrooxy-2-butanone and 3-methyl-3-nitrooxy-2-butanone, it can be observed that photolysis frequencies increase with the substitution of the alkyl chain. From these kinetic data, it can be concluded that photolysis frequencies of α- and



β-nitrooxy ketones are much higher than those obtained when considering the sum of the photolysis frequencies
for monofunctional species (see Table 2).

Products formed by the photolysis of the carbonyl nitrates were investigated by FTIR spectrometry. For both
compounds, only PAN was detected. To calculate its formation yield, concentration of PAN was plotted as a
function of $-\Delta[\text{nitrate}]_{\text{photolysis}}$, i.e., the carbonyl nitrate loss rate due to photolysis. This one was calculated by
subtracting the loss rate measured during the dark period before the photolysis to the one measured during the
photolysis. Because the yield was calculated as the initial slope of the plot, it was considered that the dark period
before irradiation was more representative than the one after. The uncertainty on the yield was calculated by
taking into account the uncertainties on the absorption cross sections of PAN (10%) and carbonyl nitrates (10%)
as well as the uncertainty on J (see table 1). For 5-nitrooxy-2-pentanone, this uncertainty is quite large because
the photolysis rate is relatively slow in comparison to loss to the reactor walls. PAN formation yields obtained
for both compounds are given in Table 1.

For 4-nitrooxy-2-butanone, PAN is formed with a yield equal to unity. Its formation can be explained by the
dissociation of the C(O)-C bond as shown in Scheme 2. This pathway also leads to the formation of the alkyl
radical $^\bullet CH_2-CH_2ONO_2$ which reacts with $O_2$ to form the corresponding peroxy radical, this latter evolving to the
formation of the alkoxy by reaction with NO. Two pathways have been considered for the evolution of the
alkoxy radical: i) the decomposition which may lead to the formation of $NO_2$ and two molecules of HCHO and
ii) the reaction with $O_2$ which produces nitrooxy ethanal, also called ethanal nitrate ($CH_2(ONO_2)$-CH(O)). None
of these two products have been detected by FTIR. However, absorption bands of ethanal nitrate are expected to
be very similar to those of the reactant and it may thus be difficult to distinguish them. In addition, ethanal
nitrate is expected to photodissociate much faster than the keto nitrate. The other photodissociation pathway is
the cleavage of the O-$NO_2$ bond. It leads to the formation of the radical $CH_3C(O)CH_2CH_2O^\bullet$ which is expected
to react with $O_2$ to form a dicarbonyl product. This product was not observed and this is in good agreement with
the formation of PAN with a yield equal to unity by the other photodissociation pathway. It should be noticed
that PAN has been detected as a primary product suggesting that its formation via the photolysis of the
dicarbonyl product is not expected. In our experiments, it was not possible to measure $NO_2$ formation yield
because large amounts of $NO_2$ (hundreds ppb) were introduced together with the carbonyl nitrate (probably due
to its decomposition during the injection).

As discussed above, since the enhancement in the cross sections is larger at the higher wavelengths, where
absorption by the nitrate chromophore is very small, it was proposed by Müller et al. (2014) that the absorption
by the carbonyl chromophore is enhanced due to the neighboring nitrate group. The authors also suggest that the
photodissociation proceeds by a dissociation of the weak O-$NO_2$ bond, i.e. that a photon absorption by one
chromophore (carbonyl group) causes dissociation in another part of the molecule (nitro group). This is not in
line with what we observed in our study. From our experiments, we conclude that the photolysis of 4-nitrooxy-2-
butanone proceeds mainly by a dissociation of the C(O)-C bond. In the former study on the photolysis of 3-
nitrooxy-2-propanone, 3-nitrooxy-2-butanone and 3-methyl-3-nitrooxy-2-butanone (Suarez-Bertoa et al., 2012),
PAN and carbonyl compounds (respectively, formaldehyde, acetaldehyde and acetone) were detected as major
products. However, branching ratio of the two pathways (dissociation of O-$NO_2$ and C(O)-C bonds) could not be
determined as the formation of these products, in particular PAN, can be explained by the two pathways.





For 5-nitrooxy-2-pentanone, formation yield of PAN has been observed to be much lower: $0.16 \pm 0.08$. As for 4-nitrooxy-2-butanone, its formation can be explained by the dissociation of the C(O)-C bond (see Scheme 3). This

result suggests that both dissociation pathways may occur and that O-NO$_2$ dissociation could be the major one. However, this was not confirmed by the detection of the dicarbonyl compound (2-oxo-pentanal) which is

expected to be formed by this pathway. Despite no standard was available for this compound, no characteristic band was observed in the residual spectrum (after subtraction of reactants and PAN spectra). Because the

photolysis rate of 5-nitrooxy-2-pentanone is very low, we suspect that the concentration of this product is below the detection limit. Nevertheless, the low PAN yield is a strong indication that O-NO$_2$ dissociation may be the

major pathway, contrary to what has been observed for 4-nitrooxy-2-butanone. This should be considered in the light of the low enhancement of absorption cross sections which has been observed for this compound. Hence, in

the case of γ-nitrooxy ketones, the enhancement of the absorption by the carbonyl chromophore has been observed to significantly decrease, leading to a lower branching ratio of the C(O)-C bond dissociation.

**3.2 OH-oxidation of carbonyl nitrates**

     Rate constants of the OH-oxidation have been measured for 4-nitrooxy-2-butanone and 5-nitrooxy-2-pentanone.

Prior to the experiments, it was checked that the carbonyl nitrates do not photolyse nor decompose/adsorb on the walls of the chamber. Figure 3 represents the kinetic plots obtained for the two carbonyl nitrates. For each

compound, several independent kinetic experiments were performed and data were combined to provide the $k_{ketonitrate}/k_{methanol}$ for each compound (see Figure 3). In order to limit errors in the quantification of reactants due

to the possible formation of carbonyl nitrates as products, only the very beginning of the experiments was taken into account for the kinetic plots. This explains the small number of experimental points. The obtained rate

constants are given in Table 3. These data are, to our knowledge, the first determinations of the rate constants for the reaction of OH with these two carbonyl nitrates. From these data, it can be concluded that 4-nitrooxy-2-

butanone and 5-nitrooxy-5-pentanone have similar reactivity towards OH radicals with rate constants equal to $(2.9 \pm 1.0) \times 10^{-12}$ and $(3.3 \pm 0.9) \times 10^{-12}$ cm$^3$ molecule$^{-1}$ s$^{-1}$, respectively.

Rate constants provided in this study, as well as those previously reported for a series of α-nitrooxy-ketones (Suarez-Bertoa et al., 2012) have been compared to those estimated using structure-activity relationships (SARs)

in Table 4. Different SARs have been evaluated: i) the one developed by Kwok and Atkinson (1995) with updated factors F(-ONO$_2$) = 0.14 and F(-C-ONO$_2$) = 0.28 from Bedjanian et al. (2018); ii) the one developed by

Kwok and Atkinson (1995) with updated factors F(-ONO$_2$) = 0.8 and F(-C-ONO$_2$) = 0.1 from Suarez-Bertoa et al. (2012); iii) the one developed by Neeb (2000) which proposes a different type of parametrization and has

been observed to be particularly accurate for oxygenated species; and iv) the one developed by Jenkin et al. (2018) which proposes a parametrization very similar to the one from Kwok and Atkinson (1995) and Bedjanian

et al. (2018). The rate constant for 5-nitrooxy-2-pentanone is reasonably well reproduced by all SARs (within a factor of 2). For 4-nitrooxy-2-butanone, only the parametrization provided by Suarez-Bertoa et al. (2012)

succeeds to reproduce the experimental value. This can be explained by the fact that this parametrization has been optimized for carbonyl nitrates while the others have been developed using the entire dataset for

compounds containing nitrate group (-ONO$_2$) (Jenkin et al., 2018) or using the dataset for alkyl nitrates (Bedjanian et al., 2018). The main difference between these parametrizations is that in Suarez-Bertoa et al.



(2012), the factor F(-ONO$_2$) is much less deactivating than for the others. This could result from electronic interactions between the two functional groups. However, Suarez-Bertoa et al. (2012) noticed that their

parametrization gives poor results for alkyl nitrates, suggesting that a specific parametrization has to be used for multifunctional species. This also suggests that the principle of SARs based on the group additivity method may

not be suitable for multifunctional molecules.

From these experiments, several oxidation products have been detected: HCHO, PAN and methylglyoxal for 4-

nitrooxy-2-butanone, and HCHO, PAN and 3-nitrooxy-propanal for 5-nitrooxy-2-pentanone. However, their quantification was highly uncertain because the infrared spectra were complex due to the presence of methanol,

isopropyl nitrite, impurities (in particular acetic acid) and their oxidation products. Dedicated mechanistic experiments should now/in the near future be performed using HONO as OH source in order to simplify the

chemical mixture.

**3.3 Atmospheric implications**

Atmospheric lifetimes of the investigated compounds have been estimated and are presented in Table 5. For the photolysis, lifetimes ($\tau_{hv} = 1/J$) were calculated for a typical actinic flux corresponding to the 1$^{st}$ July at noon and

at 40° latitude North. Under these irradiation conditions, they range from 5 to 13 hours. For OH-oxidation, lifetimes ($\tau_{OH} = (1/(k_{OH}[OH]))$) were calculated using typical OH concentrations of $2 \times 10^6$ molecule cm$^{-3}$

(Atkinson and Arey, 2003). They are both equal to approximatively two days. Hence, it appears that for 4-nitrooxy-2-butanone and for 5-nitrooxy-2-pentanone, photolysis is a more efficient sink than the oxidation by

OH radicals. Same conclusion was obtained for α-nitrooxy carbonyls (Suarez-Bertoa et al., 2012; Barnes et al., 1993; Zhu et al., 1991). However, OH-initiated oxidation is not negligible, especially under polluted conditions

where OH concentrations can be higher than $1 \times 10^7$ molecule cm$^{-3}$.

In order to evaluate the impact of these carbonyl nitrates on the nitrogen budget and the transport of NOx, it is

crucial to determine whether their atmospheric sinks, here mainly photolysis, release NO$_2$ or not. For 4-nitrooxy-2-butanone, we observed that the photolysis proceeds mainly by a dissociation of the C(O)-C bond which does

not necessarily lead to the release of NO$_2$ (see Scheme 2). In our experimental conditions (i.e., with high NO$_2$ mixing ratios), this pathway leads to the formation of PAN which was detected with a yield equal to unity.

Under more realistic NO/NO$_2$ ratio, this reaction may also produce HCHO and CO$_2$. The co-products of PAN, which could not be detected in our study, are expected to be formaldehyde + NO$_2$ or ethanal nitrate. One NO$_2$

molecule is hence released in the first hypothesis. Ethanal nitrate may react and undergo photolysis even faster than nitrooxy ketones and may thus lead to NO$_2$ release quite rapidly. However, as data on the reactivity of

ethanal nitrate are not available in the literature, one cannot provide a definite conclusion. In the case of 5-nitrooxy-2-pentanone, the dissociation of the C(O)-C bond has been observed to be a minor pathway suggesting

that the major one, which was not directly observed here, is the O-NO$_2$ dissociation. This process certainly leads to the release of NO$_2$.

**4 Conclusions**



This work represents the first study on the atmospheric reactivity of 4-nitrooxy-2-butanone and 5-nitrooxy-2-pentanone. Thanks to experiments in simulation chambers, photolysis frequencies and rate constants of the OH-oxidation were measured for the first time. From these results, it is concluded that, similarly to α-nitrooxy ketones, β-nitrooxy ketones have enhanced UV absorption cross sections and quantum yields equal or close to unity, making photolysis a very efficient sink for these compounds. For 5-nitrooxy-2-pentanone which is a γ-nitrooxy ketone, a lower enhancement of cross sections was observed leading to slightly longer atmospheric lifetimes (10-15 hours). This can easily be explained by the increasing distance between the two chromophore groups. Some photolysis products were also detected allowing estimating the branching ratio between the two possible pathways, i.e., the dissociation of the C(O)-C bond and the one of the O-NO$_2$ bond. For 4-nitrooxy-2-butanone, we conclude that the photolysis proceeds mainly by a dissociation of the C(O)-C bond which does not necessarily lead to the release of NO$_2$. In the case of 5-nitrooxy-2-pentanone, our results suggest that the dissociation of the O-NO$_2$ bond, leading to NO$_2$ release, is the major pathway. Reactivity of 4-nitrooxy-2-butanone and 5-nitrooxy-2-pentanone with OH radicals was also investigated. Both compounds have similar reactivity towards OH radicals leading to lifetimes of approximatively two days. Experimental rate constants are in good agreement with those estimated by the SAR proposed by Kwok and Atkinson (1995) when using the parametrization proposed by Suarez-Bertoa et al. (2012) for carbonyl nitrates. However, this specific parametrization does not allow reproducing experimental data for monofunctional alkyl nitrates, suggesting that specific parametrization should be used for multifunctional species. Finally, these compounds are expected to be removed from the atmosphere fairly rapidly and to act as (only) temporary reservoirs of NOx. If formed during the night, they could however contribute to longer range transport of NOx.

*Author contributions*

BPV coordinated the research project. BPV, RSB and JFD designed the experiments in simulation chambers. RSB performed the experiments with the technical support of MC and EP. RSB and MDa performed the organic syntheses. BPV, RSB and MDu performed the data treatment and interpretation. BPV and RSB wrote the paper and BPV was in charge of its final version. All coauthors revised the manuscript content, giving final approval of the version to be submitted.

*Competing interests*

The authors declare that they have no conflict of interest.

*Acknowledgments*

This work was supported by the French National Agency for Research (Project ONCEM-ANR-12-BS06-0017-01) and by the European Union within the 7th Framework Program, section "Support for Research Infrastructure-Integrated Infrastructure Initiative" through EUROCHAMP-2 project (RII3-CT-2009-228335) and the Horizon 2020 Research and Innovation Program through the EUROCHAMP-2020 Infrastructure Activity under grant agreement no. 730997.



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


**Figure, scheme and table captions**

**Scheme 1.** Photolysis pathways of 4-nitrooxy-2-butanone

        **Scheme 2.** Photolysis pathways of 5-nitrooxy-2-pentanone

**Figure 1**. Kinetic plots for *(a)* the photolysis of 4-nitrooxy-2-butanone (experiment 2) and *(b)* the photolysis of 5-nitrooxy-2-pentanone (experiment 5).

**Figure 2.** Kinetic plots for the oxidation by OH radicals of 4-nitrooxy-2-butanone and 5-nitrooxy-2-pentanone. For 5-nitrooxy-2-pentanone, data have been shifted by 0.2 in y axis.

**Table 1.** Photolysis rate and PAN yield for 4-nitrooxy-2-butanone and 5-nitrooxy-2-pentanone

        **Table 2.** Comparison of experimental photolysis rates of carbonyl nitrates with calculated ones.

**Table 3.** Rate constants for the OH-oxidation of 4-nitrooxy-2-butanone and 5-nitrooxy-2-pentanone

        **Table 4.** Comparison of experimental rate constants for the OH-oxidation of carbonyl nitrates with those
estimated by SARs

        **Table 5.** Atmospheric lifetimes of carbonyl nitrates towards photolysis and reaction with OH radicals



**Scheme 1.**





**Scheme 2.**





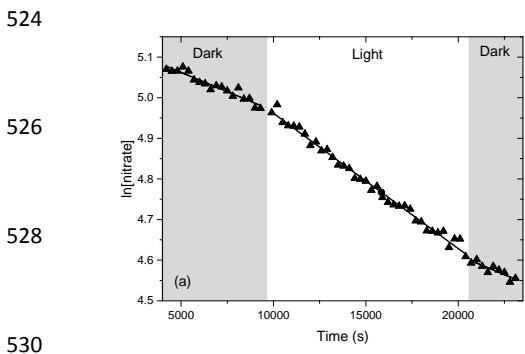 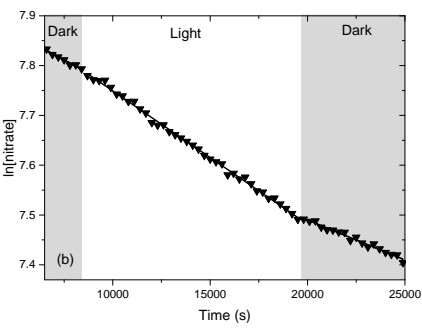

**Figure 1.**





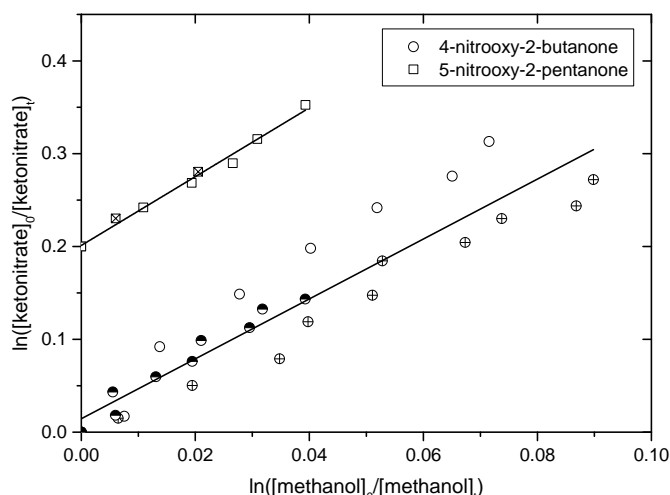

**Figure 2.**





**Table 1.**

| Compound | Experim. | $k_{before}$ [a] ($\times 10^{-5}$ s$^{-1}$) | $k_{after}$ [b] ($\times 10^{-5}$ s$^{-1}$) | (k+J) [c] ($\times 10^{-5}$ s$^{-1}$) | J [d] ($\times 10^{-5}$ s$^{-1}$) | PAN yield (%) |
|---|---|---|---|---|---|---|
| 4-nitrooxy-2-butanone | 1 | 0.8 ± 0.1 | - | 2.1 ± 0.1 | 1.3 ± 0.2 | 100 ± 35 |
| | 2 | 1.9 ± 0.2 | 2.1 ± 0.1 | 3.3 ± 0.1 | 1.3 ± 0.3 | 100 ± 40 |
| | Average | | | | 1.3 ± 0.2 | 100 ± 30 |
| | Corrected average [e] | | | | **4.2 ± 0.6** | |
| 5-nitrooxy-2-pentanone | 3 | 2.0 ± 0.2 | 1.1 ± 0.2 | 2.3 ± 0.1 | 0.7 ± 0.4 | 13 ± 9 |
| | 4 | 1.9 ± 0.2 | 1.1 ± 0.1 | 2.2 ± 0.1 | 0.7 ± 0.4 | 13 ± 9 |
| | 5 | 2.1 ± 0.2 | 1.6 ± 0.2 | 2.7 ± 0.1 | 0.8 ± 0.3 | 23 ± 13 |
| | Average | | | | 0.7 ± 0.2 | 16 ± 8 |
| | Corrected average [e] | | | | **2.2 ± 0.7** | |

[a] and [b] dark decay due to wall loss, before and after irradiation; [c] decay during irradiation period;
[d] photolysis rate; [e] atmospheric photolysis rate obtained by correcting the averaged photolysis

rate with a factor of 3.2 (see sect. 2.1).





**Table 2.**

| Compound | $J_{exp}$ ($\times 10^{-5}$ s$^{-1}$) | $J_{calc}$ ($\times 10^{-5}$ s$^{-1}$) ($\phi=1$) | J ketone + J nitrate ($\times 10^{-5}$ s$^{-1}$)[c] |
|---|---|---|---|
| **3-nitrooxy-2-propanone** <br> O <br> ONO$_2$ | 4.8 ± 0.3 <br> (Suarez-Bertoa et al., 2012) | 4.3[b] | 0.3 |
| **4-nitrooxy-2-butanone** <br> O <br> ONO$_2$ | 4.2 ± 0.6 <br> (This work) | 4.3[a] | 0.8 |
| **5-nitrooxy-2-pentanone** <br> O <br> ONO$_2$ | 2.2 ± 0.7 <br> (This work) | 5.6[a] | ND |
| **3-nitrooxy-2-butanone** <br> O <br> ONO$_2$ | 5.7 ± 0.3 <br> (Suarez-Bertoa et al., 2012) | 6.4[b] | 0.8 |
| **3-methyl-3-nitrooxy-2-butanone** <br> O <br> ONO$_2$ | 7.4 ± 0.2 <br> (Suarez-Bertoa et al., 2012) | ND | ND |

[a] calculated with estimated cross sections (see text); [b] calculated with experimental cross sections

from literature; [c] obtained from TUV NCAR model.



**Table 3.**

| Compound | $k_{nitrate}/k_{methanol}$ | $k_{nitrate} \times 10^{-12}$ (cm$^3$ molecule$^{-1}$ s$^{-1}$) |
|---|---|---|
| 4-nitrooxy-2-butanone | $3.25 \pm 0.47$ | $2.9 \pm 1.0$ |
| 5-nitrooxy-2-pentanone | $3.70 \pm 0.28$ | $3.3 \pm 0.9$ |





**Table 4.**

| Compound | $k_{exp}$ x $10^{-13}$ | $k_{SAR\ Atkinson/Bedjanian}$[1] x $10^{-13}$ | $k_{SAR\ Atkinson/Suarez}$[2] x $10^{-13}$ | $k_{SAR\ Neeb}$[3] x $10^{-13}$ | $k_{SAR\ Jenkin}$[4] x $10^{-13}$ |
|---|---|---|---|---|---|
| 3-nitrooxy-2-propanone | 6.7[5] | 2.0 | 6.6 | 5.8 | 2.5 |
| 3-nitrooxy-2-butanone | 10.1[5] | 4.5 | 13.2 | 6.8 | 3.7 |
| 3-methyl-3nitrooxy-2-butanone | 2.6[5] | 4.0 | 2.1 | 2.4 | 4.3 |
| 4-nitrooxy-2-butanone | 29[6] | 8.1 | 30.9 | 8.7 | 8.1 |
| 5-nitrooxy-2-pentanone | 33[6] | 21.4 | 22.5 | 47.6 | 19.5 |

Rate constants are expressed in cm[3] molecule[-1] s[-1]; [1] SAR developed by Kwok and Atkinson (1995) with F(-
ONO$_2$) and F(-C-ONO$_2$) from Bedjanian et al., 2018; [2] SAR developed by Kwok and Atkinson (1995) with F(-
ONO$_2$) and F(-C-ONO$_2$) from Suarez-Bertoa et al., (2012) ;[3] SAR developed by Neeb (2000); [4] SAR developed
by Jenkin et al. (2018) ; [5] experimental data from Suarez-Bertoa et al. (2012); [6] This work.



**Table 5.**

| Compound | $J \times 10^{-5}$ (s$^{-1}$) | $\tau_{h\nu}$ (hours) | $k_{OH} \times 10^{-12}$ (cm$^3$ molecule$^{-1}$ s$^{-1}$) | $\tau_{OH}$[1] (hours) |
|---|---|---|---|---|
| 4-nitrooxy-2-butanone | $4.2 \pm 0.6$ | 7 | $2.9 \pm 1.0$ | 48 |
| 5-nitrooxy-2-pentanone | $2.2 \pm 0.7$ | 13 | $3.3 \pm 0.9$ | 42 |

[1]: estimated for $[OH] = 2 \times 10^6$ molecule cm$^{-3}$