# Peer review of "Photolysis and oxidation by OH radicals of two carbonyl"

_Atmospheric Chemistry and Physics, 2019_

## Referee Comment (RC1) · Anonymous Referee #1 · 26 Sep 2019

This paper addresses the photolysis and reaction with OH of two ketonitrates for which no data were available yet. The two compounds - a $\beta$-ketonitrate and a $\gamma$-ketonitrate - were chosen primarily because there were reasons to believe that they might behave differently from $\alpha$-ketonitrates (for which experimental data is available) owing to the larger distance between the ketone and nitrate functional groups in those molecules. Previous work has suggested that the proximity of the two groups in $\alpha$-ketonitrates causes interactions which greatly enhance the photolysis rates. The present laboratory study shows that the same might apply to $\beta$-ketonitrates, but not (or not as much) to

the $\gamma$-ketonitrates. In addition, a product study has been conducted, with interesting, albeit puzzling, results. The OH-reaction rates of the two compounds are determined through a relative rate determination study, which suggests a relatively low reactivity for both compounds with OH.

The experimental methodology appears appropriate, and the analysis is convincing, although some specific points could be improved or clarified (see further below). The conclusions from this work are very relevant for our understanding of the role of VOC emissions on atmospheric chemistry, and more especially regarding their impact on nitrogen oxides. I recommend this work for publication, after the following minor comments below are addressed.

**Minor comments**

l. 37 Organic nitrates are also formed in relatively pristine locations, where they play an important role (e.g. over forests)!

l. 106 'Because of their importance for atmospheric chemistry': This is a bit odd, a robust synthesis process would be needed even if those 2 compounds were unimportant. They are model compounds. Best to drop that part of the sentence.

l. 128 What filter is being used? Wang et al. refers to different widths.

l. 135 Note that 200 DU is a very low value. This is not important for NO2, but it will be for the organic nitrates.

l. 144-145 Cyclohexane is more reactive than the ketonitrates, yes, but **not** 'very much' as stated here (7E-12 for cyclohexane, 3E-12 for the ketonitrates). The 95% OH scavenging efficiency might be a bit overestimated, at least for the experiments with $\sim$1 ppm ketonitrate.

Section 3.1 first paragraph: I see a lot of repetitions between this part and Section 2.2. It could be shortened.

Section 3.1 Why not use TUV to calculate the photolysis rates in laboratory conditions, with the known lamp spectrum? The discussion would be more convincing and straightforward. The scaling by a factor 3.2 is an approximation. It is not valid for the monofunctional nitrates of Table 2, and might be on the high side for the ketonitrates, because the lamp irradiates too much below ∼ 310 nm. I have the feeling that the 200 Dobson Units assumed here for TUV calculations (a very low ozone column, unrealistic for mid-latitudes) were chosen to mitigate the issue. Note that this remark does not affect the main conclusions of the study.

Figure 1. What is the zero of the x axis? Why not show the evolution of ln(nitrates) from the start?

Table 1. I'm surprised by the very low uncertainty estimate for $k_{before}$ (0.2) for the second compound. The corresponding dark period on the graph looks very short.

l. 248-249 For 5-nitrooxy-2-pentanone, the discrepancy between the calculated J and the experimental value could be due to the cross sections, or the quantum yields, or both. I would be therefore more cautious when discussing the enhancement of the cross section. I admit that the proposed reasoning seems plausible. We expect less cross section enhancement due to the larger distance, it is therefore likely to explain the lower J.

l. 303 and l. 372 Same thing, the low enhancement of absorption cross sections has not been observed, only assumed.

**Technical corrections**

l. 26 Replace 'increasing' by 'larger'

l. 27 Replace 'products' by 'product'

l. 33 Replace 'that' by 'than'

l. 68 Replace 'works are' by 'work is'

l. 75 Insert 'photolysis' before 'quantum yield', and drop 'photolysis' after 'unity'

l. 85 Insert 'of' before 'carbonyl nitrates'

l. 90 Replace 'process' by 'processes of'

l. 90 Insert a comma before 'leading to'

l. 107 Delete 'the' before 'Kames'

l. 138 Replace 'proportional' by 'proportionality'

l. 145 Insert 'the' before 'ketonitrates'

l. 145 Replace 'section results' by 'Results section'

l. 154 and elsewhere in the manuscript: Do not italicize 'ln' as well as 'nitrate' in equations

l. 191 Delete the ""

l. 250 Replace 'annihilated' by 'absent'

l. 263 Insert 'infrared' before 'absorption cross sections'

l. 268 You mean Scheme 1, not Scheme 2

l. 281 Delete 'together'

l. 298 Insert 'the fact that' after 'Despite'

l. 309 You mean Figure 2, not Figure 3

l. 328 Replace 'to reproduce' by 'in reproducing'

l. 330 Insert 'a' before 'nitrate group'

l. 349 Delete 'the' before 'oxidation'

l. 350 Replace 'Same conclusion' by 'An identical conclusion'

l. 378 You may drop 'leading to NO2 release'

---

## Author Comment (AC1) · 15 Oct 2019

Dear Editor, dear referee, Please find attached our answer to the referee's comments with an updated version of the manuscript which takes into account all comments. Best Regards. B. Picquet-Varrault

Please also note the supplement to this comment: https://www.atmos-chem-phys-discuss.net/acp-2019-801/acp-2019-801-AC1-supplement.zip

---

## Referee Comment (RC2) · Anonymous Referee #2 · 19 Oct 2019

This article presents results from simulation chamber studies of the photolysis and hydroxyl radical initiated oxidation of two carbonyl nitrates, 4-nitrooxy-2-butanone and 5-nitrooxy-2-pentanone. Experimental photolysis frequencies were determined and used to calculate photolysis rates and lifetimes under atmospheric conditions. Rate coefficients for the reaction of these compounds with hydroxyl radicals were also determined using the relative rate method. Some information on the products arising from these atmospheric degradation processes is reported.

Overall, this is a good piece of work, which is both relevant and of interest to the atmo-

spheric science community. The experiments have been carefully performed and the data has been treated and interpreted well. The work is suitable for publication in Atmospheric Chemistry and Physics provided that the one minor comment and technical corrections are addressed appropriately.

Minor Comments

1. The authors have measured the photolytic decay but then assume a quantum yield of unity to allow calculation of photolysis rate coefficients under atmospheric conditions. However, since it appears that the wavelength-dependent light flux in the chamber and the absorption cross-section data are both known, the effective quantum yield can in fact be determined from the ratio J(experimental) to J(maximum), where the latter term is calculated using a quantum yield of unity (Clifford et al., 2011). It is recommend that the authors do this as it will allow for a good estimate of the quantum yield for photolysis of the compounds over the range of their atmospheric absorption.

Technical Corrections

1. Lines 20-22: It is probably better to report the measured photolytic rate coefficients as j(carbonyl nitrate)/j(NO2) values, as in Clifford et al (2011). This would allow users of the data to calculate j(carbonyl nitrate) under a range of sunlight conditions.

2. Lines 22-23. The specific atmospheric conditions that the photolytic lifetimes were calculated for should be stated.

3. Line 35: No need for the abbreviation "ONs" as it does not seem to appear in the rest of the article.

4. Line 62: Give full name for PANs

5. Lines 66 and 68: Replace "works" with investigations or studies.

6. Line 83: Replace "photochemical" with atmospheric.

7. Line 90: The term "fairly fast" is not specific. This sentence should be improved.

8. Page 5 and elsewhere: The rate coefficient terms, k and J, should be in italics.

9. Line 209: The first sentence of this paragraph is not needed here.

10. Line 228: These are not "experimental" photolysis frequencies, but are in fact estimated atmospheric photolysis frequencies under specific light conditions.

11. Line 258: Define PAN

12. Line 295: Should be Scheme 2.

13. Page 15: Captions for the figures, table and schemes should be more detailed to allow the reader to view and understand them without referring to the text too much.

14. Page 17, Scheme 2: Why is decomposition of the nitrooxy radical not considered here, but it is in Scheme 1?

15. Page 19, Figure 2: What do the different data points represent? More detail should be provided in the caption and/or the figure itself.

16. Page 20, Table 1: The inclusion of the calculated atmospheric photolysis rate coefficient in this table is a bit confusing. It should be removed.

17. Page 21, Table 2: As explained in point 10 above, the use of the term J(experimental) here is wrong.

18. Page 23, Table 4: Use the times symbol instead of "x".

References:

Clifford, G. M.; Hadj-Aissa, A.; Healy, R. M.; Mellouki, A.; Muñoz, A.; Wirtz, K.; Martín-Reviejo, M.; Borrás Garcia, E.; Wenger, J. C.: The atmospheric photolysis of o-tolualdehyde, Environmental Science & Technology, 45, 9649-9657 (2011).

---

## Author Comment (AC2) · 28 Oct 2019

Dear Editor,

Please find attached our answer to Referee#2 comments. The authors would like to thank the anonymous referee for its constructive comments, corrections and suggestions that ensued. We have carefully replied to all of them and the paper has been improved following his recommendations.

Best Regards. B. Picquet-Varrault

Please also note the supplement to this comment:
https://www.atmos-chem-phys-discuss.net/acp-2019-801/acp-2019-801-AC2-supplement.zip

---

## Author Response (AR1)

Dear Editor,

First, we would like to thank you for having accepted being co-editor for our paper. Please find below the answers to referee's comments and the new version of the manuscript with corrections highlighted in blue. All comments have carefully been taken into account and corrections have been performed in the manuscript accordingly.

Best Regards.

Bénédicte Picquet-Varrault and co-authors

**Answer to Anonymous Referee #1**

First of all, the authors would like to thank the anonymous referee for this interactive discussion and its constructive comments, corrections and suggestions that ensued. We have carefully replied to all its comments and the paper has been improved following its recommendations.

All technical corrections suggested by the referee have been carefully performed. Answers have also been provided for all comments and changes have been performed accordingly. Please find below the answers to the minor comments:

*l. 37 Organic nitrates are also formed in relatively pristine locations, where they play an important role (e.g. over forests)!*
The sentence has been modified in order to be less restrictive.

*l. 106 'Because of their importance for atmospheric chemistry': This is a bit odd, a robust synthesis process would be needed even if those 2 compounds were unimportant. They are model compounds. Best to drop that part of the sentence.*
This part of the sentence has been removed.

*l. 128 What filter is being used? Wang et al. refers to different widths.*
mm Pyrex filters have been used. This information has been added in the manuscript and a Figure showing the spectrum of the lamps has been added in SI.

*l. 135 Note that 200 DU is a very low value. This is not important for $NO_2$, but it will be for the organic nitrates.*
This was a typographic error. Determination of photolysis frequencies, in particular those given in Table 2, were performed with 300 DU.

*l. 144-145 Cyclohexane is more reactive than the ketonitrates, yes, but not 'very much' as stated here (7E-12 for cyclohexane, 3E-12 for the ketonitrates). The 95% OH scavenging efficiency might be a bit overestimated, at least for the experiments with 1ppm ketonitrate.*
We agree that for experiments which were performed with 1-2 ppm of ketonitrate, only approx. 80% of the OH were scavenged. This has been changed in the manuscript.

*Section 3.1 first paragraph: I see a lot of repetitions between this part and Section 2.2. It could be shortened.*
Section 3.1 has been shortened as suggested.

*Section 3.1 Why not use TUV to calculate the photolysis rates in laboratory conditions, with the known lamp spectrum? The discussion would be more convincing and straightforward. The*

*scaling by a factor 3.2 is an approximation. It is not valid for the monofunctional nitrates of Table 2, and might be on the high side for the ketonitrates, because the lamp irradiates too much below 310 nm. I have the feeling that the 200 Dobson Units assumed here for TUV calculations (a very low ozone column, unrealistic for mid-latitudes) were chosen to mitigate the issue. Note that this remark does not affect the main conclusions of the study.*

The authors thank the referee for this very good suggestion and fully agree with it. Changes have been made accordingly: photolysis frequencies have been calculated for experimental conditions, i.e. using CESAM actinic flux. These new values are now compared to experimental data in Table 2. As expected, experimental and calculated photolysis frequencies are in good agreement for 4-nitrooxy-butanone when using the enhancement factor ($r_{nk}$) obtained for 3-nitrooxy-2-propanone. However, for 5-nitrooxy-pentannone, the experimental value is significantly lower than the calculated one confirming that the enhancement effect is reduced.

In the previous version of the manuscript, these data were also compared to the sum of the J for the monofunctional species using TUV model (for typical solar irradiation conditions and using 300 DU for the ozone total column, and not 200 DU as indicated by error in the manuscript). However, as we could not find in the literature the quantum yields of these species, this comparison was not possible for CESAM irradiation conditions and has been removed.

***Figure 1. What is the zero of the x axis? Why not show the evolution of ln(nitrates) from the start?***

The evolution of ln(nitrates) was not shown from the start (i.e. from the injection) in order to optimize the scale of the y axis. In addition, only experimental points which were taken into account for the linear regression were shown. Indeed, as explained in section 2.2, it was observed that the dark decay rate before irradiation may be significantly higher than the one after, suggesting a passivation effect of the walls. So, in order to determine wall loss decays which are as representative as possible of the one during the irradiation period, the first experimental points (after the injection of the carbonyl nitrate) were not taken into account for the linear regression leading to the determination of $k_{before}$. It has been explained more explicitly in the manuscript (l.166-171).

Following the referee comment, the figure 1 has been changed in order to show the experimental points from the start. The scale of the y axis is now optimized thanks to a break.

***Table 1. I'm surprised by the very low uncertainty estimate for $k_{before}$ (0.2) for the second compound. The corresponding dark period on the graph looks very short.***

As explained above, the first experimental points were not taken into account for the linear regression leading to the determination of $k_{before}$ because we believe that, due to passivation effect, the period just after the injection of the carbonyl nitrate (and during which the decay rate is slightly faster) may not be representative of the wall loss rate during the irradiation period. The linear regression has hence been performed for a period of 30 min before the irradiation and the value of 0.2 for the uncertainty is calculated as twice the standard deviation on these points.

*l. 248-249 For 5-nitrooxy-2-pentanone, the discrepancy between the calculated J and the experimental value could be due to the cross sections, or the quantum yields, or both. I would be therefore more cautious when discussing the enhancement of the cross section. I admit that the proposed reasoning seems plausible. We expect less cross section enhancement due to the larger distance, it is therefore likely to explain the lower J.*

*l. 303 and l. 372 Same thing, the low enhancement of absorption cross sections has not been observed, only assumed.*

The sentences have been changed.

**Additional correction made by the authors:**

In addition to the changes performed to take into account the referee's comments, we noticed that we made an error in the $J_{NO2}$ value measured in CESAM chamber. The good value which was measured at the period during which these experiments were conducted, is $2.2 \times 10^{-3}$ s$^{-1}$ and not $3.0 \times 10^{-3}$ s$^{-1}$. This has been corrected in the new version of the manuscript and calculations have been modified accordingly.

**Answer to Anonymous Referee #2**

First of all, the authors would like to thank the anonymous referee for this interactive discussion and its constructive comments, corrections and suggestions that ensued. We have carefully replied to all its comments and the paper has been improved following its recommendations.

All technical corrections suggested by the referee have been carefully performed. Answer has also been provided for the minor comment. Please find below the answers for each comment:

**Minor Comments**

*1. The authors have measured the photolytic decay but then assume a quantum yield of unity to allow calculation of photolysis rate coefficients under atmospheric conditions. However, since it appears that the wavelength-dependent light flux in the chamber and the absorption cross-section data are both known, the effective quantum yield can in fact be determined from the ratio J(experimental) to J(maximum), where the latter term is calculated using a quantum yield of unity (Clifford et al., 2011). It is recommended that the authors do this as it will allow for a good estimate of the quantum yield for photolysis of the compounds over the range of their atmospheric absorption.*

In fact, the absorption cross sections of the studied compounds are not known and were not measured in this study. They were estimated by assuming the fact that the enhancement factor $r_{nk}$ is the same as the one calculated for 3-nitrooxy-2-propanone. However, this hypothesis could not be confirmed as absorption cross sections are not available. So effective quantum yields cannot be determined from the ratio $J_{experimental}/J_{max}$.

**Technical Corrections**

*1. Lines 20-22: It is probably better to report the measured photolytic rate coefficients as j(carbonyl nitrate)/j(NO2) values, as in Clifford et al (2011). This would allow users of the data to calculate j(carbonyl nitrate) under a range of sunlight conditions.*
As proposed by the reviewer, *j(carbonyl nitrate)/j(NO2) values have been provided.*

*2. Lines 22-23. The specific atmospheric conditions that the photolytic lifetimes were calculated for should be stated.*
The irradiation conditions used to estimate the photolysis frequencies are already given in the abstract. It is not clear for us which additional information the reviewer would like us to provide.

*3. Line 35: No need for the abbreviation "ONs" as it does not seem to appear in the rest of the article*
The abbreviation has been removed.

***4. Line 62: Give full name for PANs***

It has been done.

***5. Lines 66 and 68: Replace "works" with investigations or studies.***

It has been done.

***6. Line 83: Replace "photochemical" with atmospheric.***

It has been done.

***7. Line 90: The term "fairly fast" is not specific. This sentence should be improved.***

It has been done.

***8. Page 5 and elsewhere: The rate coefficient terms, k and J, should be in italics.***

It has been done.

***9. Line 209: The first sentence of this paragraph is not needed here.***

It has been removed.

***10. Line 228: These are not "experimental" photolysis frequencies, but are in fact estimated atmospheric photolysis frequencies under specific light conditions.***

In reply to referee #1 and referee #2 comments, photolysis frequencies estimated for typical tropospheric irradiation conditions have been removed from Table 1 and 2. Only experimental values measured in the simulation chamber are presented and compared.

***11. Line 258: Define PAN***

It has been done.

***12. Line 295: Should be Scheme 2.***

It has been corrected.

***13. Page 15: Captions for the figures, table and schemes should be more detailed to allow the reader to view and understand them without referring to the text too much.***

We did our best to improve the captions of the figures, tables and schemes. We hope they are understandable now.

***14. Page 17, Scheme 2: Why is decomposition of the nitrooxy radical not considered here, but it is in Scheme 1?***

This is an omission and the oxidation scheme has been completed accordingly. However, from the state of the art on the alkoxy chemistry, we expect the decomposition channel to be negligible in comparison to the reaction with $O_2$.

**15. Page 19, Figure 2: What do the different data points represent? More detail should be provided in the caption and/or the figure itself.**

The different symbols represent different experiments. This has been added in the figure caption.

**16. Page 20, Table 1: The inclusion of the calculated atmospheric photolysis rate coefficient in this table is a bit confusing. It should be removed.**

We fully agree with this comment. Photolysis frequencies estimated for typical tropospheric irradiation conditions have therefore been removed from Table 1 and 2.

**17. Page 21, Table 2: As explained in point 10 above, the use of the term J(experimental) here is wrong.**

See answer to comment 10.

**18. Page 23, Table 4: Use the times symbol instead of "x".**

It has been done.

**New version of the manuscript with changes highlighted (in blue)**

[revised manuscript text omitted]

---

## Author Response (AR2)

Dear Andreas Hofzumahaus,

We would like to thank you for your constructive comments and we have taken all of them into account.
We have corrected the following points:

- When it was relevant, we have replaced "photolysis rates" by "photolysis frequencies". We agree that it was improperly used in some cases.
- We have replaced "J" by "j" to express the photolysis frequency.
- The specific atmospheric conditions used to calculate the atmospheric photolysis frequency (albedo and total ozone column) were indeed not indicated in the abstract, although they were given in the section 3.3. So we have added them in the abstract. We have also given the aerosol and cloud optical depth which were 0.235 and 0, respectively (values proposed by default values in TUV model).

Best Regards.

Bénédicte Picquet-Varrault